# The innate immune sensor IFI16 recognizes foreign DNA in the nucleus by scanning along the duplex

Sarah A Stratmann[1†], Seamus R Morrone[2†], Antoine M van Oijen[1,3*], Jungsan Sohn[2*]

[1]University of Groningen, Groningen, Netherlands; [2]Johns Hopkins University School of Medicine, Baltimore, United States; [3]University of Wollongong, Wollongong, Australia

**Abstract** The ability to recognize foreign double-stranded (ds)DNA of pathogenic origin in the intracellular environment is an essential defense mechanism of the human innate immune system. However, the molecular mechanisms underlying distinction between foreign DNA and host genomic material inside the nucleus are not understood. By combining biochemical assays and single-molecule techniques, we show that the nuclear innate immune sensor IFI16 one-dimensionally tracks long stretches of exposed foreign dsDNA to assemble into supramolecular signaling platforms. We also demonstrate that nucleosomes represent barriers that prevent IFI16 from targeting host DNA by directly interfering with these one-dimensional movements. This unique scanning-assisted assembly mechanism allows IFI16 to distinguish friend from foe and assemble into oligomers efficiently and selectively on foreign DNA.

**\*For correspondence:** vanoijen@uow.edu.au (AMvO); jsohn@jhmi.edu (JS)

[†]These authors contributed equally to this work

## Introduction

The host innate immune system detects infection by directly recognizing molecular signatures associated with pathogens (*Janeway and Medzhitov, 2002*; *Medzhitov and Janeway, 2000*). Remarkably, such signatures include universal building blocks of all life, such as DNA and RNA (*Janeway and Medzhitov, 2002*; *Orzalli and Knipe, 2014*; *Paludan and Bowie, 2013*). In the cytoplasm, the immune system relies on the absence of endogenous DNA, and thus marks all detected DNA as 'foreign' (nonself) (*Orzalli and Knipe, 2014*; *Paludan and Bowie, 2013*). However, DNA viruses often evade the cytosolic detection machineries, as their genomes are not exposed until reaching the nucleus (*Orzalli and Knipe, 2014*; *Paludan and Bowie, 2013*). The host counters this infection strategy in the nucleus by directly assembling supramolecular signaling platforms that trigger inflammatory responses on invading foreign DNA, but not on its own genomic material (*Johnson et al., 2013*; *Li et al., 2012*; *Kerur et al., 2011*). Although key players that target foreign dsDNA in the host nucleus have been identified (*Orzalli and Knipe, 2014*; *Paludan and Bowie, 2013*), the molecular mechanisms by which these sensors distinguish self from nonself dsDNA remain unknown.

The interferon-inducible protein 16 (IFI16) is a key innate immune sensor that detects foreign dsDNA and uses it as a scaffold to assemble supramolecular signaling platforms in both the host nucleus and cytoplasm (*Unterholzner et al., 2010*; *Johnson et al., 2013*; *Li et al., 2012*; *Kerur et al., 2011*; *Orzalli et al., 2012*) (*Figure 1A*). IFI16 plays a central role in defense against a number of pathogens (e.g herpes simplex virus-1) (*Unterholzner et al., 2010*; *Johnson et al., 2013*; *Li et al., 2012*; *Kerur et al., 2011*; *Orzalli et al., 2012*). On the other hand, persistent IFI16 signaling is associated with autoimmunity (e.g. Sjögren's syndrome) (*Mondini et al., 2007*;

**eLife digest** The immune system defends us from attacks by viruses, bacteria and other microbes. One way that the immune system can identify these invaders is by detecting genetic material from the microbes. Like us, many of these microbes have genetic material made of a two-stranded molecule called DNA. A protein called IFI16 is an important sensor in immune cells that can bind to foreign DNA, but not to the DNA of the host. If the ability to discriminate between host and foreign DNA is lost, then immune cells start to attack the body's own tissues, which can lead to severe "autoimmune" diseases. However, it is not clear how IFI16 and other sensor proteins are able to tell foreign and host DNA apart.

DNA in host cells is packaged in particular proteins to form structures called nucleosomes that can make it difficult for other proteins to bind to the DNA. When foreign DNA enters the cell, IFI16 promotes the formation of nucleosomes on these molecules, but the nucleosomes are further apart than those in host DNA, which leaves longer stretches of "exposed" DNA between the nucleosomes. In 2014, a group of researchers reported that multiple molecules of IFI16 associate with each other and form clusters on exposed foreign DNA. Here, Stratmann, Morrone et al. – including some of the researchers from the earlier work – investigated how IFI16 selectively binds to foreign DNA using a combination of biochemical and single-molecule techniques.

The experiments show that to start with, a few molecules of IFI16 bind to stretches of exposed foreign DNA and then move along the DNA strands. As more molecules of IFI16 bind to this DNA, they bump into each other and start to form clusters that eventually become immobile. Further experiments show that the more tightly packed nucleosomes in host DNA molecules act as a barrier to IFI16 cluster formation because they interfere with the ability of the protein to move along the DNA.

Stratmann, Morrone et al.'s findings show that IFI16 can only trigger immune responses when it binds to stretches of exposed DNA that are long enough to allow the assembly of IFI16 clusters. The next challenge will be to see whether other DNA sensors employ similar strategies to detect foreign DNA.

Choubey et al., 2010; Mondini et al., 2010; Gugliesi et al., 2013; Smith and Jefferies, 2014). The molecular mechanisms by which IFI16 selectively targets foreign dsDNA remain unknown. To establish a functional signaling platform, IFI16 must overcome two challenges. First, individual IFI16 molecules must be able to locate one another on large pathogen genomes with sizes ranging from $10^5$ to $10^6$ base pairs (bps). Second and more importantly, this assembly mechanism can only take place on foreign dsDNA and must be inhibited on host dsDNA (*Figure 1A*). Here, we report the observation of a unifying molecular mechanism that explains how IFI16 resolves these central issues in initiating its foreign-dsDNA sensing pathways.

## Results and discussion

To identify the mechanisms underlying assembly of IFI16 signaling platforms on DNA, we monitored the oligomerization kinetics of FRET donor and acceptor labeled IFI16 on naked dsDNA (FRET: fluorescence resonance energy transfer; *Figure 1B* and *Figure 1—figure supplement 1*). Previous work demonstrated the existence of such oligomers and reported on their equilibrium binding properties but did not provide insights into the assembly mechanisms (*Morrone et al., 2014*). Using various dsDNA fragment sizes present in excess, we observe that the assembly rate increased non-linearly and by 50-fold from 60 to 200 bps dsDNA, above which it stayed constant (up to 600 bps; *Figure 1B,C*). With a dsDNA-binding footprint of ~15 bp for one IFI16 (*Morrone et al., 2014*), our results indicate that about 4 copies are required to initiate assembly, and about 10 IFI16 molecules are required for optimal oligomeric assembly (*Figure 1C*). Further, the assembly rate constants scaled linearly with the IFI16 concentration for all measured DNA lengths (*Figure 1—figure supplement 1* and *Supplementary file 1A*), indicating that a purely cooperative assembly mechanism is unlikely. In line with this observation, previous work reported relatively small contributions of

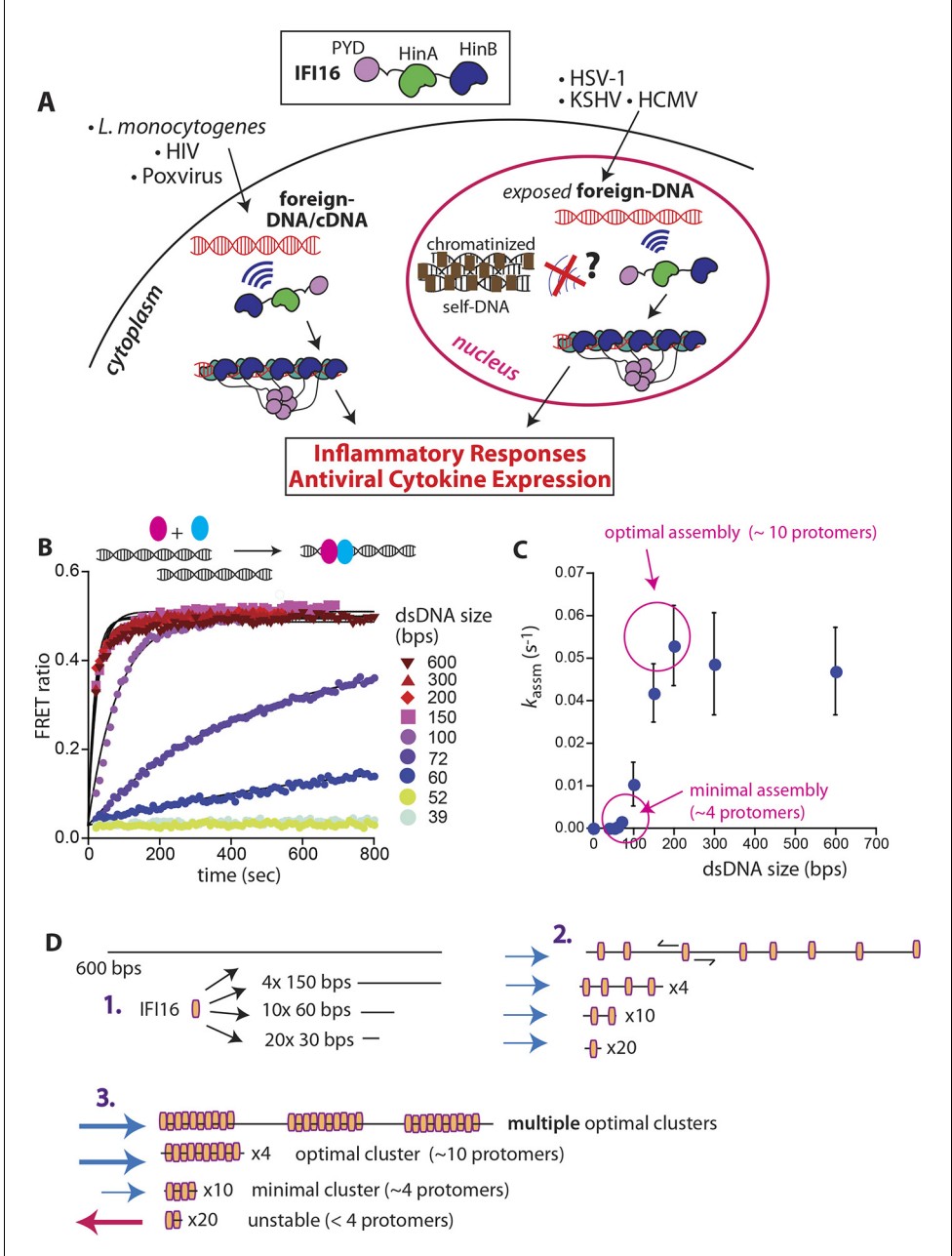

**Figure 1.** IFI16 assembles faster on longer dsDNA. (A) Top: IFI16 is composed of three functional domains flanked by unstructured linkers, namely one pyrin domain (PYD) and two dsDNA-binding Hin domains (HinA and HinB; Hin: hematopoietic interferon-inducible nuclear antigen). Bottom: IFI16 detects foreign dsDNA from invading pathogens in both the host nucleus and cytoplasm. (B) Top: a cartoon scheme for FRET experiments. The two differentially colored ovals represent fluorescently (Dylight-550 and Dylight-650) labeled IFI16. Bottom: The time-dependent changes in the emission ratio between FRET donor and acceptor labeled IFI16 (50 nM) were monitored at 33 μg/ml of each dsDNA (e.g. sixfold higher than the dissociation constant for 39-bp dsDNA [*Morrone et al., 2014*]). Lines are fits to a first-order exponential equation (see *Figure 1—figure supplement 1* for 25 nM protein). All shown representative experiments were performed at least three times. (C) A plot of observed assembly rates ($k_{assm}$s) vs. dsDNA-sizes (see also *Figure 1—figure supplement 1*). (D) 1D-diffusion assisted assembly mechanism can explain the observed assembly profile of IFI16. 1. At the same mass-concentrations, the number of individual dsDNA fragments present in each assay is inversely proportional to the length of dsDNA. 2. Individual IFI16 molecules initially bind dsDNA at random positions and diffuse one-dimensionally while searching for other respective protomers; the number of IFI16 residing on the same dsDNA fragment should be proportional to the length of dsDNA (e.g. there are four times more individual 150-bp fragments than 600-bp fragments) 3. IFI16 fails to assemble into an oligomer on dsDNA shorter than 60 bp (indicated by a red arrow pointing left). The saturating rates can be explained if the final FRET signals arise from formation of distinct optimal oligomers.

*Figure 1 continued on next page*

*Figure 1 continued*

The following figure supplement is available for figure 1:

**Figure supplement 1.** FRET assembly assays using 25 nM donor and acceptor labeled IFI16 compared to 50 nM in *Figure 1B*.

cooperativity in oligomerization with Hill constants near 2 for DNA substrates up to 2000 bp (*Morrone et al., 2014*).

Our ensemble-averaged, solution-phase observations of the faster assembly on longer dsDNA suggest a model in which IFI16 scans along dsDNA to increase the probability of encountering other IFI16 molecules (*Figure 1D*). To directly test such a mechanism, we used single-molecule fluorescence imaging to track the movements of individual Cy5-labeled IFI16 molecules on stretched, double-sided attached λ-phage dsDNA (λdsDNA; 48.5 kbps) (*Figure 2A* and *Video 1*). *Figure 2B* shows that individual IFI16 molecules one-dimensionally (1D) diffuse on λdsDNA while bound for several seconds. The diffusion coefficient of IFI16 increased with ionic strength, indicating that IFI16 does not maintain a continuous electrostatic interaction with the dsDNA backbone, but instead moves along the λdsDNA scaffold by executing microscopically small steps (*Blainey et al., 2006*) (*Figure 2B* and *Figure 2—figure supplement 1*).

An IFI16 construct lacking the oligomerizing pyrin domain (PYD) (IFI16$^{HinAB}$; see also *Figure 1A*) showed similar diffusional properties, suggesting that the dsDNA-binding HIN200 domains are responsible for 1D diffusion. Upon applying higher concentrations of IFI16 with a constant supply of proteins into our flow cell, we observed a gradual formation of distinct, immobile clusters along λdsDNA (*Figure 2C*, *Figure 2—figure supplement 2*, and *Video 2*). Over time, we also observed an increase in the number of molecules per cluster and a concomitant decrease in the diffusion coefficient (*Figures 2C,D*). We analyzed the impact of flow on the diffusion coefficient and diffusion bias and found it to be not significant for cluster formation (*Figure 2—figure supplement 4*). Single-molecule intensity analysis revealed that the lower limit of the number of IFI16 molecules found in immobile clusters is equivalent to eight protomers (*Figure 2D* and *Figure 2—figure supplement 3*), which also corroborate the optimal complex (ten protomers) suggested from *Figure 1C*. Furthermore, individual IFI16 molecules either formed new clusters or joined other clusters in a stochastic manner, and the immobile clusters formed faster with higher IFI16 concentrations (*Figure 2E*). The rate of addition of IFI16 molecules to clusters is independent on the size of the existing cluster, confirming the absence of strong cooperativity in assembly (*Figure 2C*; bottom panel).

The 1D diffusion of IFI16 on dsDNA explains why the assembly rates increase with the DNA length in the bulk experiments (*Figure 1C*). With the longer dsDNA acting as an antenna, it allows binding of more IFI16 while 1D diffusion facilitates dynamic association (*Figure 1D*). The saturation of the assembly rate (*Figure 1C*) can be

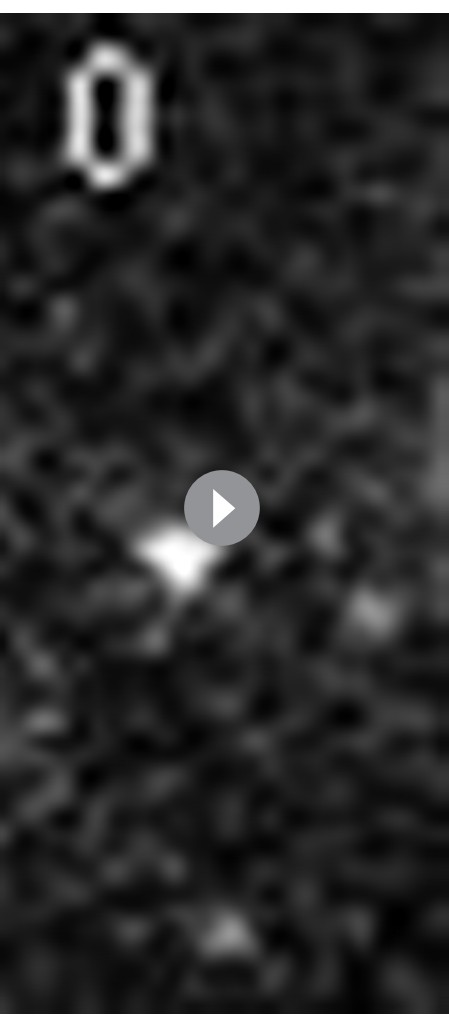

**Video 1.** Single Cy5-labeled IFI16 protein (1 nM) moving on double-biotinylated dsλ-DNA at 300 mM KCl (without flow). The movie is played at 5x acceleration. This video is related to *Figure 2*.

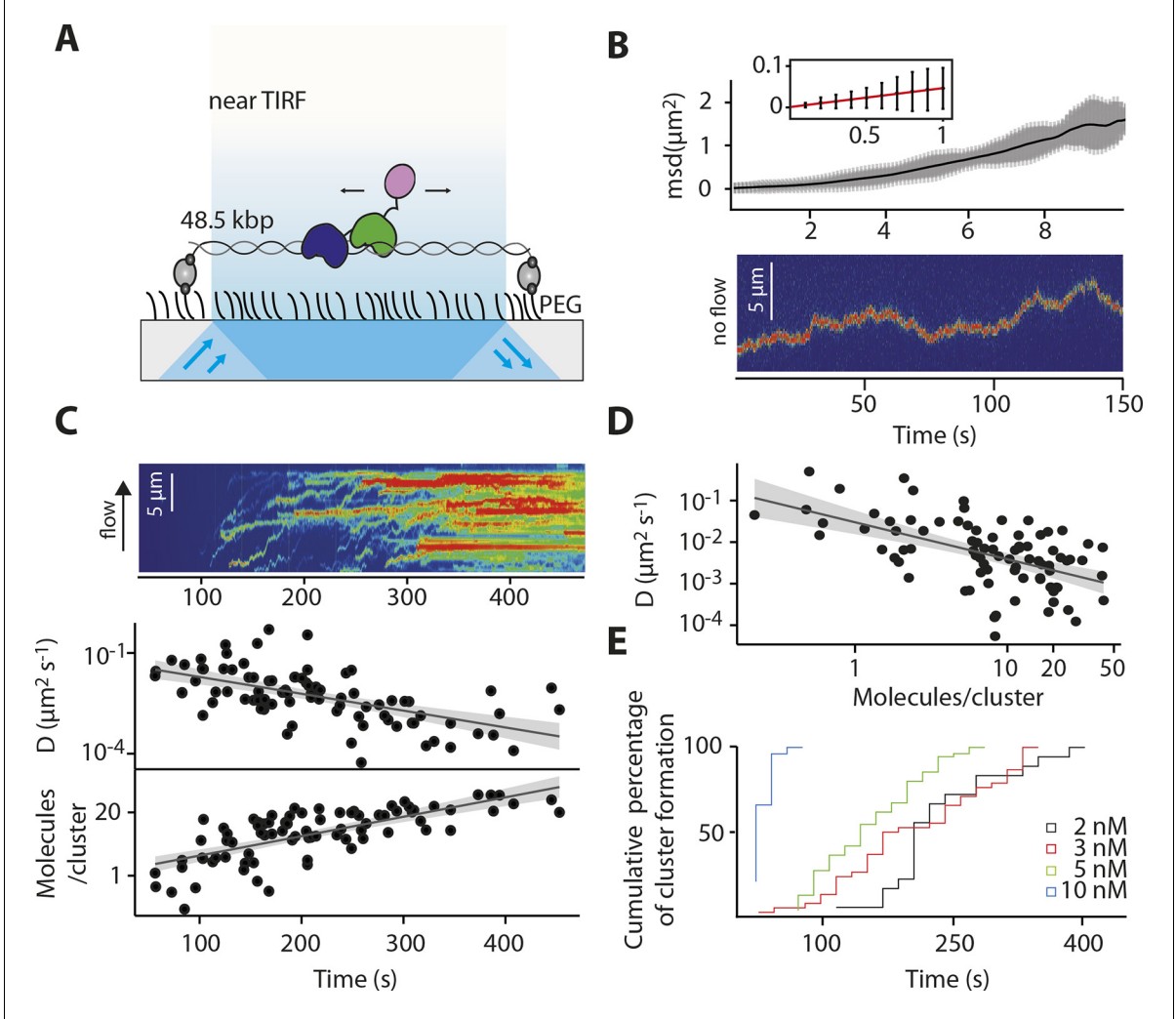

**Figure 2.** IFI16 scans dsDNA. (**A**) Illustration of the TIRF setup. λdsDNA is anchored to the pegylated coverslip surface by either one or two biotinylated oligonucleotide linkers. Single-biotinylated λdsDNA is constantly stretched by flow during measurements and used for the clustering and nucleosome experiments, whereas double-biotinylated λdsDNA is stably attached whilst being stretched and used without flow for single-molecule diffusion coefficient analysis. (**B**) Double-biotinylated λdsDNA is anchored to the surface and Cy5-labeled IFI16 molecules (800 pM) are visualized in near TIRF while bound to DNA. Top: Mean-square displacement (msd) trajectories are fitted within their linear regime to calculate the 1D-diffusion coefficient. Bottom: A sample kymograph of a single molecule stably bound for tens of second to DNA and exerting Brownian motion. (**C**) Elevated IFI16 concentrations result in clustering along λdsDNA. Top: A sample kymograph of multiple IFI16 molecules (3 nM) diffusing along λdsDNA. IFI16 molecules display a net motion along the flow direction. Bottom: Time-resolved clustering is accompanied by a decrease in diffusion coefficients and increase in the number of molecules per cluster. (**D**) The diffusion coefficient inversely correlates with the number of IFI16 per cluster, resulting in immobile, stable oligomers. (**E**) Cluster formation is IFI16-concentration dependent.

The following figure supplements are available for figure 2:

**Figure supplement 1.** Dependence of the diffusion coefficient of IFI16 and IFI16HinAB mutant on salt concentration.

**Figure supplement 2.** (Left) Representative fluorescence images of IFI16-Cy5 on DNA molecules: Concentrations of 1 and 5 nM result in the resolution of single IFI16 molecules and distinguishable Ifi16 clusters, respectively, whereas a higher concentration of 10 nM eventually leads to DNA congestion with protein clusters or filaments.

**Figure supplement 3.** Fluorescence data of 5 nM IFI16 are converted to cluster sizes (see *Figure 2—figure supplement 2*).

**Figure supplement 4.** Impact of the flow-induced drift on diffusion and clustering is minimal.

explained by the square dependence of the diffu-
sional search time on length: at a sufficiently long
dsDNA length, the dissociation rate of an individ-
ual IFI16 will be faster than the time needed to
scan along the entire length of the DNA. In addi-
tion, longer DNA substrates work no longer as
antennae, but as traps, since individual IFI16 mol-
ecules are farther apart and thus less likely to
encounter one another (*Hu et al., 2006*;
*Turkin et al., 2015*; *2016*). Overall, the results of
our bulk and single-molecule experiments are

**Video 2.** Cy5-labeled IFI16 molecules (3 nM) moving
and clustering on single-biotinylated dsλ-DNA at 160
mM KCl at constant flow from left to right. This video is
related to *Figure 2*.

consistent with the dsDNA-size dependent binding in vitro(*Morrone et al., 2014*), which also corre-
lates with the IFI16-induced inflammatory responses in vivo(*Unterholzner et al., 2010*). Thus, we
propose that the 1D-diffusion mediated assembly plays a key role in regulating the overall IFI16-
mediated immune responses.

It has long been speculated that chromatinization acts as the key feature that allows IFI16 to dis-
tinguish host from foreign DNA in the nucleus (*Kerur et al., 2011*; *Unterholzner and Bowie, 2011*;
*Li et al., 2012*; *Orzalli et al., 2012*; *2013*; *Johnson et at., 2014*); IFI16 oligomerizes on exposed
invading foreign-dsDNA before it becomes hetero-chromatinized. Previous in vivo work demon-
strated that transfected chromatinized SV40 DNA is able to evade IFI16 oligomerization and down-
stream responses (*Orzalli et al., 2013*). Nevertheless, the molecular mechanism by which IFI16 could
use chromatinization to distinguish self from nonself has yet to be identified. To directly address this
issue, we first used a competition-binding assay to investigate how IFI16 interacts with dsDNA frag-
ments containing two nucleosomes with varying spacer sizes (6, 30, 50, and 70 bps; *Figure 3A* and
*Figure 3—figure supplement 1*). Here, di-nucleosomes with 6-, 30-, and 50-bp spacer failed to com-
pete against IFI16-bound FAM-labeled 70-bp dsDNA, (*Figure 3B*). On the other hand, the di-nucleo-
some with 70-bp spacer competed similarly as 70-bp naked dsDNA, but significantly more weakly
than naked 300-bp dsDNA (*Figure 3B*). In FRET assembly assays, di-nucleosomes with spacers
shorter than 70-bp failed to support assembly (*Figure 3C*), consistent with our FRET kinetics assays
using naked dsDNA (*Figure 1B*). The 70-bp spacer di-nucleosome supported oligomerization of
IFI16; however, the assembly kinetics was again similar to that of naked 70-bp dsDNA, but not that
of naked 300-bp dsDNA (*Figure 3C*). Taken together, these results show that efficient IFI16 cluster
formation requires a minimal length of 50-70 base pairs of exposed dsDNA. Considering that the
size of dsDNA linker between two nucleosomes is about 20 to 30 bps in mammals (*McGhee et al.,
1983*), these results directly support the hypothesis that chromatinization is a key deterrent for pre-
venting the assembly of IFI16 signaling platforms on self-dsDNA.

To test whether the inhibitory effect of chromatin directly arises by interfering with the 1D diffu-
sion of IFI16, we visualized the movement of individual IFI16 molecules on DNA with nucleosomes.
We introduced randomly localized nucleosomes in λdsDNA using recombinant human histone
octamers and tagged nucleosome positions with fluorescent antibodies against the N-terminal tail of
histone H4 (*Figure 4—figure supplement 1*). The application of hydrodynamic flow resulted in the
single IFI16 molecules being pushed to one direction and clustering at nucleosomal sites on
λdsDNA, unable to overcome the octamers by diffusion (*Figure 4A*, *Figure 4—figure supplement
2*, and *Video 3*). Without the antibody, the motion of IFI16 was still confined, whereas for bare
λdsDNA, IFI16 moved with a high processivity along the entire strand (*Figures 4B,C*, and *Figure 4—
figure supplement 3*). These observations are consistent with the bulk experiments (*Figure 3*), and
confirm that nucleosomes directly restrict the 1D diffusion of IFI16 and consequently limit the assem-
bly of IFI16 signaling platforms on dsDNA.

The molecular mechanism by which innate immune sensors distinguish self from foreign dsDNA in
the host nucleus has been a major unresolved question in innate immunology (*Kerur et al., 2011*;
*Unterholzner and Bowie, 2011*; *Li et al., 2012*; *2013*; *Orzalli et al., 2012*; *2013*; *Johnson et al.,
2014*). The oligomerization of IFI16 on under-chromatinized foreign DNA plays a key role not only in
initiating inflammatory and antiviral responses, (*Monroe et al., 2014*; *Orzalli et al., 2012*;
*Kerur et al., 2011*), but also in regulating the hetero-chromatinization and silencing of viral dsDNA
(*Johnson et al., 2014*; *Orzalli et al., 2013*). By using time-resolved bulk and single-molecule fluores-
cence assays, we demonstrate here that IFI16 ID scans along exposed dsDNA to assemble into

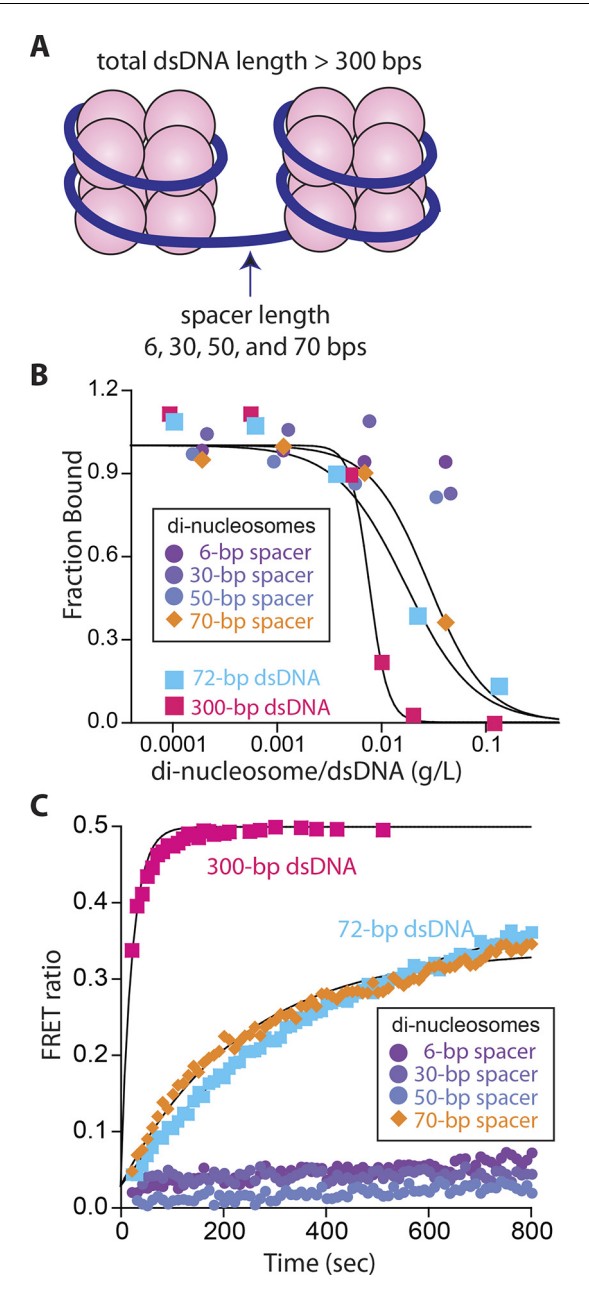

**Figure 3.** Nucleosomes inhibit oligomerization. (**A**) A cartoon of di-nucleosome constructs with varying dsDNA-spacers. (**B**) Competition binding assays using IFI16 bound FAM-labeled 70-bp dsDNA against various di-nucleosomes and naked dsDNA. The lines are fits to: $1/(1+([DNA_{competitor}]/IC_{50})^{Hill\ constant})$, where $IC_{50}$ indicates the concentration of competitor at 50% efficiency. The mass-concentration of each competitor was calculated using dsDNA, but not histones. (**C**) The time-dependent changes in the emission ratio between the FRET donor and acceptor labeled IFI16 (50 nM) were monitored at 33 µg/ml of each nucleosome or naked dsDNA. The lines are fits to a first-order exponential equation.

The following figure supplement is available for figure 3:

**Figure supplement 1.** Agarose gels with nucleosome preparations.

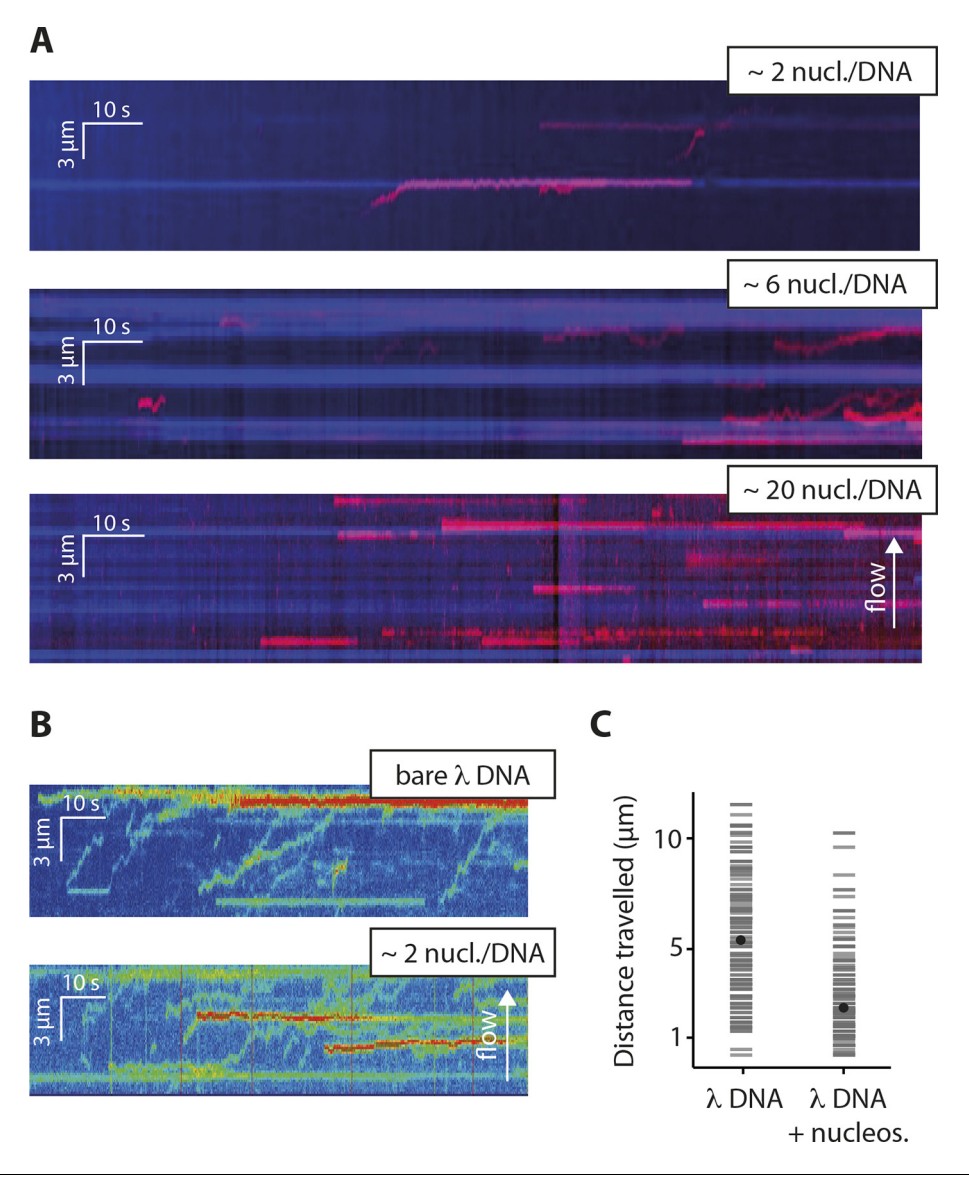

**Figure 4.** Nucleosomes inhibit 1D-diffusion. (**A**) Kymographs of Cy5-labeled IFI16 (magenta) binding to λdsDNA with varying numbers of nucleosomes tagged with anti-H4-Atto488 (blue). The number of nucleosomes per λdsDNA were estimated by quantifying IFI16 clustering sites for the lowest nucleosome/λdsDNA ratio (*Figure 4—figure supplement 3C*), yielding ~2 nucleosomes/ λdsDNA. At low nucleosome concentrations (~2 to 6 nucleosomes/DNA), IFI16 binds to λdsDNA and diffuses with the flow direction, until encountering a nucleosome. At higher concentrations (~20 nucleosomes per λdsDNA), individual IFI16 show only very short diffusive movements upon binding. (**B**) On naked λdsDNA, IFI16 travels with the flow to the free tip (top), whereas it oligomerizes along the path on nucleosome-loaded λdsDNA (bottom). (**C**) The overall travel distance of single IFI16 on nucleosome-loaded λdsDNA is reduced compared to bare λdsDNA (nucleosomal λdsDNA: $N = 167$, bare λdsDNA: $N = 141$).

The following figure supplements are available for figure 4:

**Figure supplement 1.** Reconstituted nucleosomes on restricted λdsDNA: EcoRI digestion generated l-DNA fragments of 21 kbp, 7.5 kbp, 5.8 kbp, 5.6 kbp, 4.8 kbp, and 3.5 kbp.

**Figure supplement 2.** Single-molecule co-localization probability of anti-H4- Atto488 with IFI16-Cy5 on nucleosomal biotin-λdsDNA and on biotin-601-nucleosomes (N = 419 and 472, respectively).

*Figure 4 continued on next page*

*Figure 4 continued*

**Figure supplement 3.** Nucleosomes interfere with the 1D-diffusion of IFI16.

distinct clusters and that chromatinization is sufficient to inhibit IFI16 from targeting host dsDNA for assembly. In vivo, this 1D scanning mechanism allows a limited number of IFI16 molecules to allocate each other on large genomes of invading pathogens. In combination with 3D sampling of binding sites on a collapsed DNA molecule, this process optimizes the oligomerization and downstream signaling time. While the clustering on dsDNA presents a tempting explanation for the role of IFI16 in viral gene silencing, future in vivo experiments await to test this. IFI16 belongs to the family of AIM2-like receptors, which include other nuclear and cytosolic foreign dsDNA-sensors. It will be interesting to determine whether and how these other related sensors use exposed dsDNA as a 1D 'digital ruler' to regulate their signaling platform assembly. This family of sensors is implicated in a number of autoimmune disorders (*Mondini et al., 2007*; *2010*; *Choubey et al., 2010*; *Gugliesi et al., 2013*; *Smith and Jefferies, 2014*); how regulation of assembly is disrupted may provide insights into these afflictions.

## Materials and methods

### Protein expression and purification
Human full-length IFI16 and IFI16HinAB were cloned and expressed using *E. coli* T7 express cells (NEB) as a C-terminally His6-tagged protein as described in Morrone et al. (*Morrone et al., 2014*).

### DNA ligand preparation
dsDNA shorter than 90-bp were obtained from Integrated DNA Technologies (IDT) as described in Morrone et al. (*Morrone et al., 2014*). The complementary strands were dissolved and mixed in 1:1 molar ratio, melted at 95°C for 10 min, and the temperature was lowered to 25°C at a rate of 1°C/min. Ligands of greater length were obtained by polymerase-chain reaction (PCR) using the Maltose Binding Protein fusion tag cloning sequence as template and primers of appropriate sequence for a final length as indicated in the paper. Plasmids containing the Widom-601/603 sequence with indicated linker lengths were a kind gift of Dr. Gregory Bowman. The nucleosomal DNA was obtained by PCR from these constructs with appropriate primers. All substrates were gel-purified.

### Fluorescent labeling
DyLight-550, DyLight-650, or Cy5 fluorophore was incorporated to IFI16 using maleimide chemistry (purchased from Thermo Scientific and Invitrogen) and was performed as described in Morrone, et al. (*Morrone et al., 2014*). The label to protein ratio was ~ 1:1. Fluorescein-labeled dsDNA72 was obtained from IDT.

### Octamer refolding and nucleosome reconstitution
Lyophylized Xenopus laevis histones H1A, H2A, H3, and H4 were a kind gift of Dr. Cynthia Wolberger. Octamer refolding and nucleosome reconstitution was performed as described in Luger et al. (*Luger et al., 1999*), at a 2:1 molar ratio of octamer:DNA. An agarose gel of

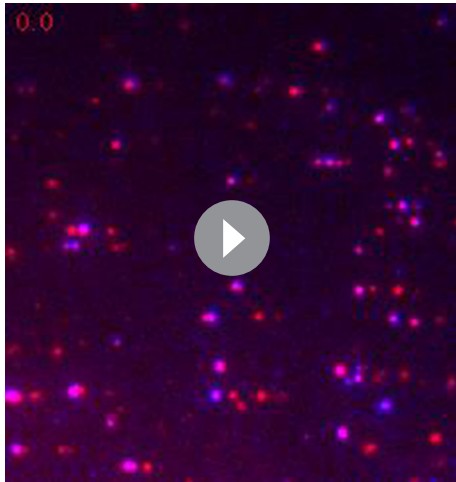

**Video 3.** Cy5-labeled IFI16 molecules (1 nM, red) moving on λ-DNA, with on average two nucleosomes per DNA, at constant flow from left to right. Nucleosomes are tagged with fluorescent anti-H4 (Atto488, blue). Most nucleosomes co-localize with IFI16, indicating that they are efficiently blocking 1D-diffusion by IFI16. This video is related to *Figure 4*.

reconstituted nucleosomes is shown in *Figure 3—figure supplement 1*.

## Bulk biochemical assays

All absorption, fluorescence anisotropy, and fluorescence excitation/emission experiments were performed in a Tecan Infinite M1000. All experiments were performed at least three times and the fits to data were generated by Kaleidagraph software (synergy).

## Competition binding assays

All reactions were performed in 40 mM HEPES pH 7.4, 160 mM KCl, 5% glycerol, 1 mM EDTA, 0.1% triton-X-100, 5 mM DTT (Reaction Buffer). Here, 300 nM IFI16 and 4.5nM fluorescein-labeled dsVACV72 were incubated together at room temperature for 20 min. Increasing concentrations of competing DNA were added to the reaction to a final concentration of 100 nM IFI16 and 1.5 nM dsVACV72, and the changes in fluorescence anisotropy were recorded as indicated in Morrone et al. (*Morrone et al., 2014*).

## FRET time dependence assays

All reactions were performed in reaction buffer. 66 µg/ml of each dsDNA or di-nucleosomes was placed in the plate wells, and the reaction was initiated by adding an equivalent volume of IFI16-550 and IFI16-650 (1:1 molar ratio) to the indicated final concentration. The dead time between addition of IFI16 and the first measurement was 15-20 s. The final dsDNA molar-concentrations are at least sixfold higher than their determined binding constants by fluorescence anisotropy assays described in Morrone, et al. (*Morrone et al., 2014*), and the FRET ratio for each time point was calculated by dividing the acceptor emission (678 nm) by the donor emission (574 nm) (*Morrone et al., 2014*).

## Fluorescence microscopy assays

Microscope coverslips (Corning) were plasma-cleaned and activated with 100 mM KOH, silanized with 3-Aminopropyl-triethoxysilane in acetone and functionalized with PEG-NHS and biotin-PEG-NHS (Laysan-Bio) in sodium bicarbonate with a 1:3 mass ratio (*Tanner and van Oijen, 2010*). Streptavidin (Sigma Aldrich) was used for anchoring the biotinylated DNA substrates to the coverslip surface. Flow channels were constructed with custom-made PDMS chips to obtain dimensions of 10 mm length, 100 µm height, and 1 mm width, and connected to a syringe pump to allow constant flow during the measurements.

Single-molecule assays were performed in 40 mM HEPES (pH 7.4), 160 mM KCl, 1 mM EDTA, 2 mM DTT, 10% glycerol, 0.1% Triton-X100, 250 µg/ml BSA, 1 mM Trolox, 40 mM glucose, 250 nM glucose oxidase, 60 nM catalase, unless otherwise stated. All assays were performed at room temperature. We applied 100 nM YoPro-1 iodide (Life Technologies) at the end of the measurements to visualize the DNA substrates.

Reactions were illuminated with a 488-nm and 641-nm laser (Coherent), and images were acquired with an EMCCD camera (Hamamatsu). We used MetaVue imaging software (Molecular Devices) for data acquisition and ImageJ and R for analysis.

## λ-DNA templates for microscopy

Lambda-DNA (New England Biolabs) was biotinylated at one or both ends by ligation of the respective complementary biotinylated oligos according to Tanner et al. (*Tanner and van Oijen, 2010*) (oligo sequences are given in *Supplementary file 1B*; purchased from IDT). Single-biotinylated DNA templates were stretched by constant flow (20 µl/min) throughout the experiments (IFI16 clustering and nucleosome assays). Double-biotinylated DNA templates were applied to the flow cell at high flow velocity (100 µl/min). This allowed binding of the DNA to two biotin moieties, while the DNA was stretched. Free DNA was washed out, IFI16 applied to the flow cell, and flow was then stopped for acquisition of the single molecule diffusion traces.

To reconstitute nucleosomes, recombinant histone H2A/H2B dimers and H3/H4 tetramers (New England Biolabs) were assembled on biotin-λ-DNA and biotin-601 sequence (Epicypher) by salt-gradient dialysis (2 M to 0.3 mM NaCl in 5 steps, each step with an incubation time of at least 2 hr) in 10 mM Tris/HCl, pH 7.4, 0.1 mM EDTA, 0.5 mM DTT. We tested nucleosome reconstitution by an EMSA assay on digested λ-DNA (*Figure 4—figure supplement 1*). For this, we generated DNA

fragments by digestion with EcoRI (NEB), purified them (Qiagen DNA spin columns) and reconstituted nucleosomes in the concentration ratios that we also used for full-length λ-DNA.

## Single molecule co-localization of IFI16 with nucleosomes tagged with antibodies

Antibodies against human histone H4 and H2B were chosen to target the N- terminal histone tails (Santa Cruz, sc-8657 and sc-8650), and labeled with Atto488-NHS (Life technologies) in PBS at pH 7.0. Labeled antibodies were negatively tested against bare DNA and Ifi16 clusters in the microscopy assay, and showed high specificity for nucleosomal DNA preparations only.

λ-DNA templates, loaded with nucleosomes and tagged with anti-H4, were constantly stretched in the flow channel. Ifi16 co-localized strongly with the nucleosome signal (*Video 3*, *Figure 3—figure supplement 1*). In contrast, biotinylated 601-sequence prepared with nucleosomes and tagged with anti-H4-Atto488, hardly showed co-localization with Ifi16, indicating, that there is not sufficient exposed dsDNA available for binding (*Figure 4—figure supplement 2*).

## Drift correction

For the clustering experiments, we applied a flow of 0.02 ml/min to our flow cell design of 0.1 mm height and 1 mm width, giving a volumetric flux of 0.33 cm/s. We expect the DNA molecules to be on average 0.2 µm away from the surface (*Tafvizi et al., 2008*), giving a velocity at this distance y of (*Tafvizi et al., 2011*):

$$v_y = \tfrac{3}{2} v_{avg} \left( \frac{hy - y^2}{h^2} \Big/ 4 \right) = 40 \; \mu m/s, \quad with \quad v_{avg} = \frac{2}{3} v_{max}$$

The Stokes drag force that acts on the DNA and bound IFI16 molecules is then described by $F = 6\pi\eta r v \left(1 + \frac{9r}{16y}\right) = 1.5 fN,$

with viscosity $\eta$, radius r and distance y.

We can define this force within the diffusion coefficient D by using the drift velocity v, calculated from single-molecule trajectories (*Leith et al., 2012*):

$$with \; v = \frac{\sum_j^{all\;traj.} x_{j,final} - x_{j,\,initial}}{\sum_j^{all\;traj.} t_{j,final} - x_{j,\,initial}} \approx 0.127 \; \mu m/s.$$

In order to evaluate the effect of flow on the clustering mechanism, we implemented a 1D-random walk Monte-Carlo simulation (*Figure 2—figure supplement 4*). Here, we calculated the expected search time for two IFI16 molecules (with D=0.026 µm²/s) to allocate each other on a λ-DNA molecule congested with a varying amount of other diffusing IFI16 molecules (10, 50, 100 molecules). As all particles are equal, we segmented the DNA according to the total number of molecules bound and calculated the effective distance between two particles by using the absolute distance modulo the segment length in order to take the periodic boundaries into account.

We allowed a maximum distance of 10 nm to consider two particles having met and dimerized. To simulate flow similar to our experimental conditions, to each random step the term *vdt* was added (The Python code is found in *Source code 1*).

## Acknowledgements

We thank Drs. Cynthia Wolberger and Gregory Bowman for their assistance in generating di-nucleosomes, Michiel Punter and Dr. Victor Krasnikov for advice with data analysis and microscopy, and Dr. Karl Duderstadt and Dr. Bastian Niebel for a critical reading of the manuscript. JS acknowledges support from the Jerome L. Greene Foundation and AvO from the European Research Council (ERC Starting 281098) and the Australian Research Council (FL140100027).

## Additional information

### Competing interests

AMvO: Reviewing editor, *eLife*. The other authors declare that no competing interests exist.

### Funding

| Funder | Author |
|---|---|
| Jerome L. Greene Foundation | Seamus Morrone<br>Jungsan Sohn |
| European Research Council | Sarah Stratmann<br>Antoine M van Oijen |
| Australian Research Council | Antoine M van Oijen |

The funders had no role in study design, data collection and interpretation, or the decision to submit the work for publication.

### Author contributions

SAS, SRM, Acquisition of data, Analysis and interpretation of data, Drafting or revising the article, Contributed unpublished essential data or reagents; AMvO, JS, Conception and design, Analysis and interpretation of data, Drafting or revising the article, Contributed unpublished essential data or reagents

## Additional files

### Supplementary files

• Supplementary file 1. (A) dsDNA-mediated oligomerization rates of FRET-labeled IFI16. (B) Oligonucleotides used in this study.

• Source code 1. Code for random walk simulation written in Python.

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
