## [Decision Letter]

Thank you for submitting your work entitled "The innate immune sensor IFI16 recognizes foreign DNA in the nucleus by scanning along the duplex" for consideration by *eLife*. Your article has been evaluated by John Kuriyan (Senior editor) and three reviewers, one of whom is a member of our Board of Reviewing Editors.

The reviewers have discussed the reviews with one another and the Reviewing editor has drafted this decision to help you prepare a revised submission.

Summary:

The manuscript reports on single molecule tracking studies of the IFI16 DNA sensor on DNA molecules without or without loaded nucleosomes. The work shows that IFI16 can diffuse along the naked DNA and form oligomers. Movement along the DNA is impeded by nucleosomes if spaced closely enough and oligomer formation is reduced. The authors conclude that IFI16 scanning is required for oligomer formation. This is a very timely and potentially important piece of work.

Essential revisions:

The reviewers felt there were a few areas that needed strengthening:

1) The connection of scanning to IFI16 function is probably beyond the scope this manuscript, but the text needs to more clearly state that IFI16 scanning on DNA could contribute to both innate signaling and epigenetic silencing initiated by IFI16 binding to DNA.

2) The reviewers all felt that the conclusion that scanning was required for oligomer formation needed further documentation or was too strong. One reviewer felt that additional biophysical studies were needed to relate scanning to oligomer formation, while the other two had questions about the specific interpretation of the results. For example, the question of how the binding stoichiometry of IFI16 relates to the two HIN DNA-binding domains was considered and what is the role of each of the HIN domains? How were the data interpreted to allow the conclusion that scanning is required for oligomer formation as opposed to a parallel process to cooperative binding? How was the lower limit of the number of IFI16 molecules per immobile cluster determined? Much of this involves revising the manuscript for a more general audience but may require some additional experiments.

3) There are some minor issues on referencing as addressed below that should be addressed.

There are a number of details that are embedded in the individual reviews so we are including those so the authors can understand the issues more fully. The areas that need to be addressed are summarized above.

*Reviewer #1:* This manuscript reports on single molecule tracking studies of the IFI16 protein on dsDNA. Consistent with previous studies from the Sohn lab, IFI16 binding to DNA requires large DNA fragments involving oligomers of IFI16. To examine the mechanism of assembly of the oligomers, they tracked single molecules of IFI16 on stretched DNA. They observed that IFI16 tracks along stretches of DNA and they interpret this as required for assembly of the oligomers. Addition of nucleosomes to the DNA at sufficiently high density prevented IFI16 from moving and oligomerizing.

The scanning of IFI16 along DNA to assemble oligomers to initiate signaling for either innate responses or epigenetic silencing is an important observation. The definition of a mechanism by which IFI16 distinguishes between foreign and self DNA is an important result as well, so this is potentially an important piece of work for the field. I have one reservation. It is not clear from the presentation how the results demonstrate that scanning is required for the assembly of the oligomers versus just happening in parallel to cooperative binding. The text needs to better define the reasoning and interpretation of the results to allow this conclusion. This may be in part due to a lack of background of this reviewer, but manuscripts for *eLife* should be written for the general reader.

For example, in the fourth paragraph of the main text it is not clear how the lower limit of the number of IFI16 found in immobile clusters was determined to be 8.

I have a few minor citation issues as well:

1) Main text, second paragraph. Johnson et al., 2013 is really derivative of the other earlier references, which should include Orzalli et al., 2012.

2) Main text, second paragraph. IFI16 is associated with autoimmunity, but it not known how it "promotes" it.

3) Main text, sixth paragraph. This states that the chromatin inhibition hypothesis has not been tested directly. Orzalli et al., 2013 did compare gene expression of SV40 DNA in chromatin versus free DNA.

*Reviewer #2:* In this manuscript, Stratmann et al. used a combination of bulk biochemistry and single molecule analysis to investigate the filament assembly process of IFI16 on dsDNA, and the mechanism by which IFI16 measures the dsDNA length as a basis for self vs. non-self-discrimination. Through FRET kinetics analysis, the authors showed that IFI16 binds to dsDNA in a length dependent manner (which was also reported in their earlier paper, Morrone et al., PNAS 2014), and drew the conclusion that 10 protomers are necessary to form the stable filament nucleus. They then used TIRF microscopy to show that IFI16 can diffuse on DNA under flow and proposed that this facilitates filament assembly and DNA length measurement. The authors also used the nucleosome coated DNA to show that IFI16 can only diffuse within the nucleosome-free region of DNA, and proposed that the nucleosome structure limits its filament assembly on self DNA.

The proposed model of DNA sensing by 1D diffusion is interesting and novel. However, I have a number of concerns. In particular, the authors failed to show the importance of diffusion in filament assembly, which is the main conclusion of this manuscript.

The length sensitivity observed between ~50 bp-200 bp in Figure 1 does not necessarily need to involve 1D diffusion. This is particularly true since the kinetic data in Figure 1 can be reasonably fit without invoking diffusion (and just by using first-order exponential equation). In order to show the importance of diffusion, the authors should show that 3D diffusion alone cannot explain the observed assembly rate, and there should be at least two parameters (1D diffusion and 3D diffusion rates) affecting the assembly rate.

Another reason that makes me doubt about the importance of diffusion is the range of DNA length that IFI16 appears to sense. The authors observed the assembly rate increases with the DNA length but plateaus around ~200 bp. Since the authors clearly showed that IFI16 can diffuse far beyond 200 bp (>2-3 kb), this observation cannot explain the observed saturation at ~200 bp. On the other hand, this can be simply explained by the requirement of ~10 protomers to form a stable filament, as suggested by the authors. Note that there are proteins that form filaments in a nucleic acid length-dependent manner but without 1D diffusion. If diffusion is important for filament assembly, there should be a length dependence beyond 200 bp, especially at low protein concentration where the 3D diffusion-mediated filament assembly is significantly compromised. Again, without kinetic modeling that clearly shows the inconsistency with 3D diffusion and requirement for 1D diffusion, it is hard to arrive at the same conclusion as the authors.

Another concern is that the 1D diffusion was shown only under flow. I understand that this is required to visualize diffusion (as you need to stretch the DNA), but this makes diffusion mostly uni-directional, and could likely affect the scanning processivity and the contribution of diffusion to filament assembly efficiency. More importantly, long DNA (perhaps those of viral DNA) collapses in the absence of flow, in which case 3D diffusion (or hopping), in comparison to 1D diffusion, could be more important for the filament assembly.

In summary, while the authors showed 1D diffusion under flow and length sensitivity in bulk biochemistry, I don't think their data suggest that 1D diffusion is required for the length sensitivity and filament assembly.

*Reviewer #3:* This is a timely and important study that addresses a fundamental question in immune response: How does a nuclear DNA sensor recognize pathogenic dsDNA from host dsDNA within the same subcellular compartment (i.e., nucleus), a recognition critical for the initiation of immune signals? The study focuses on the nuclear DNA sensor IFI16, the interferon inducible protein 16. The authors used fluorescence assays to investigate the binding of IFI16 to foreign dsDNA, first measuring the number of IFI16 molecules necessary for assembly. Using single-molecule fluorescence imaging, the authors go on to monitor the diffusion of IFI16 along lambda-phage dsDNA. Lastly, as chromatinization was previously proposed as a critical contributor to the ability of IFI16 to distinguish between host and foreign dsDNA, the authors perform competition-binding assays and time-resolved fluorescent assays to gain further insight into the impact of chromatinization on pathogenic DNA sensing. Overall, this study elegantly incorporates fluorescence imaging assays to address current questions relevant to the initiation of intrinsic and innate immune responses. This study can have a broad impact on understanding mechanisms of host defense against nuclear-replicating DNA viruses.

1) Based on the FRET results shown in Figure 1, the authors propose that four molecules of IFI16 are required for the initiation of the assembly, while ten molecules are required for oligomerization. This calculation relies on the important previous finding from this lab that the footprint of one IFI16 is ~15 bp (Morrone et al. 2014). The authors should clarify if this requires the binding of both HIN domains and if so, how they ensured in this new experiment that both HIN domains of IFI16 are simultaneously bound to the dsDNA. Can there be a circumstance in which a HIN-A domain of one IFI16 molecule and the HIN-B of another molecule bind to dsDNA? Would this interfere with the calculation of the necessary footprint?

2) The authors propose that IFI16 scans along dsDNA (Figure 1 and Figure 2). However, the mechanism of scanning is still not well defined. IFI16 has two HIN domains that can bind dsDNA with different efficiency. Is the scanning along the DNA substrate accomplished in a similar manner as a two-headed molecular motor? Are both HIN domains required for scanning? Should one of the HIN domains release its binding for the scanning to occur? Some of these questions are difficult to address and may be outside the scope of the current paper. However, one question that could be addressed and that would provide important mechanistic insight is how do the HIN-A and HIN-B domains behave individually? The authors could test this by performing the same studies with individual HIN-A or HIN-B domains. If the diffusion and dissociation behaviors of the individual domains change relative to the HIN-A/HIN-B construct, this may suggest that both domains are required for scanning.

3) The authors show beautifully the movement of IFI16 along the lambda-phage dsDNA and propose that this diffusion is critical for IFI16-mediated immune responses. However, it is not clear if this movement is a property specifically required for triggering immune response rather than a property involved in the transcriptional regulatory functions of IFI16. Are the immobile clusters sufficient for triggering immune response? The authors could add to their Discussion to address these possibilities.

---

## [Author Response]

*The reviewers felt there were a few areas that needed strengthening:*

*1) The connection of scanning to IFI16 function is probably beyond the scope this manuscript, but the text needs to more clearly state that IFI16 scanning on DNA could contribute to both innate signaling and epigenetic silencing initiated by IFI16 binding to DNA.*

Our results indicate that one-dimensional diffusion allows IFI16 to assemble into distinct clusters specifically on naked or underchromatized DNA of several kbp length. These clusters are seen as the active “firing” form, because the oligomerized PYD domains are thought to be the platform for downstream signaling (e.g. [1, 2]). In the revised manuscript, we expanded the Discussion to highlight this aspect.

We are aware of the recent literature on epigenetic silencing that might be initiated by IFI16 as well (e.g. [3]). It is difficult, however, with our in vitroassays to derive a possible epigenetic silencing mechanism mediated by IFI16. However, we revised the discussion in our revised manuscript to raise this possibility.

*2) The reviewers all felt that the conclusion that scanning was required for oligomer formation needed further documentation or was too strong. One reviewer felt that additional biophysical studies were needed to relate scanning to oligomer formation, while the other two had questions about the specific interpretation of the results. For example, the question of how the binding stoichiometry of IFI16 relates to the two HIN DNA-binding domains was considered and what is the role of each of the HIN domains? How were the data interpreted to allow the conclusion that scanning is required for oligomer formation as opposed to a parallel process to cooperative binding? How was the lower limit of the number of IFI16 molecules per immobile cluster determined? Much of this involves revising the manuscript for a more general audience but may require some additional experiments.*

These are all important questions and we are grateful for the reviewers to have raised them. Below, we will discuss these issues one by one. In summary, we obtained assembly rate constants as a function of protein concentration to exclude a major contribution of cooperativity, and introduced new single-molecule analyses to support the same assertion. We calculated the effect of flow on the one-dimensional diffusion of individual IFI16 proteins and show it to be negligible. Finally, we inserted additional information and explanation in our manuscript to better address a number of the issues raised by the reviewers.

*3) There are some minor issues on referencing as addressed below that should be addressed. There are a number of details that are embedded in the individual reviews so we are including those so the authors can understand the issues more fully. The areas that need to be addressed are summarized above.*

Further detailed below.

Reviewer #1:

*This manuscript reports on single molecule tracking studies of the IFI16 protein on dsDNA. Consistent with previous studies from the Sohn lab, IFI16 binding to DNA requires large DNA fragments involving oligomers of IFI16. To examine the mechanism of assembly of the oligomers, they tracked single molecules of IFI16 on stretched DNA. They observed that IFI16 tracks along stretches of DNA and they interpret this as required for assembly of the oligomers. Addition of nucleosomes to the DNA at sufficiently high density prevented IFI16 from moving and oligomerizing. The scanning of IFI16 along DNA to assemble oligomers to initiate signaling for either innate responses or epigenetic silencing is an important observation. The definition of a mechanism by which IFI16 distinguishes between foreign and self DNA is an important result as well, so this is potentially an important piece of work for the field. I have one reservation. It is not clear from the presentation how the results demonstrate that scanning is required for the assembly of the oligomers versus just happening in parallel to cooperative binding. The text needs to better define the reasoning and interpretation of the results to allow this conclusion. This may be in part due to a lack of background of this reviewer, but manuscripts for eLife should be written for the general reader.*

The reviewer raises a very important issue, one that was also brought up by reviewer #2. To address this concern, we performed experiments in which we use solution-based FRET to monitor the assembly rate of IFI16 aggregates on DNA for different protein concentrations.

As shown in [Supplementary-material SD1-data], for all DNA lengths, a doubling of the concentration results in a doubling of rates, a result that cannot be reconciled with any major cooperativity in protein concentration.

Furthermore, as shown in Author response table 1 below, the calculated Hill constants from our previous FRET assays that monitor the equilibrium assembly of IFI16 oligomers are generally below two for dsDNA sizes ranging from 60 to 2000 bps (Morrone et al., PNAS 2014, Figure 4 and Author response table 1) [4]. These observations consistently disagree with a highly-cooperative assembly model where the expected Hill constants should be near the number of molecules per cluster (~ ten). Overall, we cannot exclude modest amounts of cooperativity (with Hill coefficients near two), but our data cannot be fit using cooperative binding curves with larger Hill coefficients.

Author response table 1. Binding of IFI16 against various DNA using FRET

DNAKFapp (nM)KFapp (normalized for footprint) (nM)Hill ConstantdsHSV60165 ± 256601.3 ± 0.2dsVACV7246 ± 162181.8 ± 0.1dsDNA12014 ± 61131.6 ± 0.2dsDNA2007.8 ± 1.71041.8 ± 0.3dsDNA3006.2 ± 0.51242.3 ± 0.5dsDNA6002.3 ± 0.4911.8 ± 0.1dsDNA20000.8 ± 0.11021.6 ± 0.2

Further, we analyzed the growth kinetics of individual IFI16 clusters on the DNA. Strong cooperativity in cluster formation would result in a strong deviation from linearity in the number of molecules per cluster over time. After all, cooperativity would allow for faster binding of additional IFI16 once the first ones are bound or, in the extreme case, would support the simultaneous binding of a number of IFI16 molecules. Figure 2 shows clearly that the rate of growth of a cluster is linear with time and independent on the number of IFI16 molecules already present.

We modified the Discussion to reflect that modest amount of cooperativity may be present, but that our observations are most consistent with the assembly model where the major mechanism of cluster formation is driven by one-dimensional diffusion.

*For example, in the fourth paragraph of the main text it is not clear how the lower limit of the number of IFI16 found in immobile clusters was determined to be 8.* We included in our manuscript a better description of how we calibrated the number of IFI16 molecules per cluster. Briefly, we divide the fluorescence intensity of a cluster by the mean intensity value of a single IFI16 protein. Since the proteins are labeled with, on average, a single dye and the single dyes give rise to the same fluorescence intensity (provided the excitation conditions are similar), the intensity values of individual proteins form a narrow distribution, whose mean value can be used to ‘count’ the number of proteins in a cluster. We obtained the mean intensity value for single IFI16 molecules by analyzing those that bind transiently to the lambda DNA, and calculated from this mean intensity value (for the same laser power, time acquisition settings, TIRF angle) the molecules in clusters that form at the critical concentration of 3 nM. We added the intensity histogram in Figure 2—figure supplement 2.

*I have a few minor citation issues as well: 1) Main text, second paragraph. Johnson et al., 2013 is really derivative of the other earlier references, which should include Orzalli et al., 2012.*

Agreed; we referenced Orzalli et al., 2012, in the revised manuscript.

2) Main text, second paragraph. IFI16 is associated with autoimmunity, but it not known how it "promotes" it.

We revised the text to make this section more clear.

*3) Main text, sixth paragraph. This states that the chromatin inhibition hypothesis has not been tested directly. Orzalli et al., 2013 did compare gene expression of SV40 DNA in chromatin versus free DNA.*

Thank you; we revised the manuscript to include a brief statement on the in vivowork.

Reviewer #2:

*[…] The proposed model of DNA sensing by 1D diffusion is interesting and novel. However, I have a number of concerns. In particular, the authors failed to show the importance of diffusion in filament assembly, which is the main conclusion of this manuscript. The length sensitivity observed between ~50 bp-200 bp in Figure 1 does not necessarily need to involve 1D diffusion. This is particularly true since the kinetic data in Figure 1 can be reasonably fit without invoking diffusion (and just by using first-order exponential equation). In order to show the importance of diffusion, the authors should show that 3D diffusion alone cannot explain the observed assembly rate, and there should be at least two parameters (1D diffusion and 3D diffusion rates) affecting the assembly rate.*

The reviewer brings up an important point that is partly related to reviewer #1’s concerns about a possible cooperativity mechanism. We can exclude cluster formation taking place via three- dimensional search mechanisms in the absence of cooperativity: With our observation that a minimum number of four IFI16 molecules are needed to form a stable cluster and in the absence of cooperativity, chances that these four molecules will arrive at the same time on the same point of DNA are negligibly small (the probability roughly scales inversely with the number of available binding sites available per protein raised to the power *N*, with *N* being the number of proteins that need to arrive simultaneously).

An alternative mechanism, however, would be a highly cooperative one (as also suggested by Reviewer #1). This is unlikely for the reasons that we described in our reply to Reviewer #1 (please see the discussion above).

*Another reason that makes me doubt about the importance of diffusion is the range of DNA length that IFI16 appears to sense. The authors observed the assembly rate increases with the DNA length but plateaus around ~200 bp. Since the authors clearly showed that IFI16 can diffuse far beyond 200 bp (>2-3 kb), this observation cannot explain the observed saturation at ~200 bp. On the other hand, this can be simply explained by the requirement of ~10 protomers to form a stable filament, as suggested by the authors. Note that there are proteins that form filaments in a nucleic acid length-dependent manner but without 1D diffusion. If diffusion is important for filament assembly, there should be a length dependence beyond 200 bp, especially at low protein concentration where the 3D diffusion-mediated filament assembly is significantly compromised. Again, without kinetic modeling that clearly shows the inconsistency with 3D diffusion and requirement for 1D diffusion, it is hard to arrive at the same conclusion as the authors.*

The reviewer brings up an important point. While it is true that a minimum of ~200 bp is needed to provide sufficient space to form a stable oligomer, it is not necessarily true that a further increase in length would further speed up the reaction kinetics. After all, lengthening DNA would also increase the average distance between neighboring proteins bound to DNA, and thus slow down their association via 1D diffusion [5]. We would like to point out two papers that we published in the last month addressing this particular issue. One of these papers [6] studies the association kinetics of two DNA-sliding entities and finds no strong DNA-length dependence. A theoretical paper [7] describes an analytical model based on a mean-field kinetic description of linear extended sinks (representing DNA) that can trap molecules present in a solution. The model presented there accurately describes the observed independence of reaction rate on DNA length and further demonstrates that the major contribution to association kinetics is one- dimensional sliding. While we agree with the reviewer that the data mentioned above alone cannot positively confirm the importance of a scanning component, it also cannot be used to exclude such a contribution.

*Another concern is that the 1D diffusion was shown only under flow. I understand that this is required to visualize diffusion (as you need to stretch the DNA), but this makes diffusion mostly uni-directional, and could likely affect the scanning processivity and the contribution of diffusion to filament assembly efficiency. More importantly, long DNA (perhaps those of viral DNA) collapses in the absence of flow, in which case 3D diffusion (or hopping), in comparison to 1D diffusion, could be more important for the filament assembly.*

We acquired the diffusion coefficients for individual IFI16 molecules and the HinAB construct on DNA templates, which were tethered at both ends to the surface and did not require a constant flow. Therefore, the robust 1D diffusion seen in Figure 2 and Figure 2—figure supplement 1 is an intrinsic feature of IFI16/HinAB.

Indeed, applying flow during the clustering experiments changes the diffusion behavior in terms of directionality and slightly in the value of the diffusion coefficient. We added a correction term for calculating the diffusion coefficient under flow (also described in [8]), and also added a Monte-Carlo model, in which we simulated the clustering behavior with and without flow bias (Figure 2—figure supplement 4, and Supplementary Information). Here we did not observe a strong effect induced by flow.

The direction bias might change the absolute timing of clustering in our experiments. However, we want to stress with Figure 2 that the property of diffusion/scanning correlates with the number of molecules within an aggregate. Large clusters are immobile elements on the DNA with a high stability that we propose would correspond with the initial platforms for downstream signaling.

As the reviewer points out, in the cell IFI16 clusters on collapsed DNA. This indeed might change clustering, as now a combination of 1D scanning and 3D hops to nearby DNA segments is favorable to find other proteins bound to DNA (also described in [9]). Having now gained knowledge about the mechanism of IFI16 binding to DNA, i.e. its ability of 1D scanning over kbp lengths, we included this aspect better in our Discussion.

Reviewer #3:

*1) Based on the FRET results shown in Figure 1, the authors propose that four molecules of IFI16 are required for the initiation of the assembly, while ten molecules are required for oligomerization. This calculation relies on the important previous finding from this lab that the footprint of one IFI16 is ~15 bp (Morrone et al. 2014). The authors should clarify if this requires the binding of both HIN domains and if so, how they ensured in this new experiment that both HIN domains of IFI16 are simultaneously bound to the dsDNA. Can there be a circumstance in which a HIN-A domain of one IFI16 molecule and the HIN-B of another molecule bind to dsDNA? Would this interfere with the calculation of the necessary footprint?*

The footprint of IFI16 was determined using multiple dsDNA sizes and full-length protein [4]. Also as shown in the previous study by Morrone et al. [4], mutants with either the HinA or the HinB domain show a similar binding behavior on short DNA substrates (72 bp) as the full-length protein, albeit with a small reduction in DNA affinity in the absence of either domain. Due to the geometry of the Hin domains and the footprint of one Hin domain being ~ 7 bp [10], we do not expect a significant change in the footprint length.

*2) The authors propose that IFI16 scans along dsDNA (Figure 1 and Figure 2). However, the mechanism of scanning is still not well defined. IFI16 has two HIN domains that can bind dsDNA with different efficiency. Is the scanning along the DNA substrate accomplished in a similar manner as a two-headed molecular motor? Are both HIN domains required for scanning? Should one of the HIN domains release its binding for the scanning to occur? Some of these questions are difficult to address and may be outside the scope of the current paper. However, one question that could be addressed and that would provide important mechanistic insight is how do the HIN-A and HIN-B domains behave individually? The authors could test this by performing the same studies with individual HIN-A or HIN-B domains. If the diffusion and dissociation behaviors of the individual domains change relative to the HIN-A/HIN-B construct, this may suggest that both domains are required for scanning.*

We agree that this aspect of Hin domain contribution to the binding is very interesting, especially in the context of a comparison with the DNA binding domain of Aim2 and its potential diffusion behavior. However, we feel that those experiments would be beyond the scope of this manuscript, which is focused on the distinction of self and foreign DNA.

*3) The authors show beautifully the movement of IFI16 along the lambda-phage dsDNA and propose that this diffusion is critical for IFI16-mediated immune responses. However, it is not clear if this movement is a property specifically required for triggering immune response rather than a property involved in the transcriptional regulatory functions of IFI16. Are the immobile clusters sufficient for triggering immune response? The authors could add to their Discussion to address these possibilities.*

We added this aspect to our conclusion and referenced the appropriate work suggesting PYD- domain dependent signaling.

*References*

1) Lu, A. et al. Unified polymerization mechanism for the assembly of ASC-dependent inflammasomes. Cell 156, 1193-206 (2014).

2) Vajjhala, P.R., Mirams, R.E. & Hill, J.M. Multiple binding sites on the pyrin domain of ASC protein allow self-association and interaction with NLRP3 protein. J Biol Chem 287, 41732-43 (2012).

3) Orzalli, M.H., Conwell, S.E., Berrios, C., DeCaprio, J.A. & Knipe, D.M. Nuclear interferon-inducible protein 16 promotes silencing of herpesviral and transfected DNA. Proc Natl Acad Sci U S A 110, E4492-501 (2013).

4) Morrone, S.R. et al. Cooperative assembly of IFI16 filaments on dsDNA provides insights into host defense strategy. Proc Natl Acad Sci USA 111, E62-71 (2014).

5) Hu, T., Grosberg, A.Y. & Shklovskii, B.I. How proteins search for their specific sites on DNA: the role of DNA conformation. Biophys J 90, 2731-44 (2006).

6) Turkin, A. et al. Speeding up biomolecular interactions by molecular sledding. Chemical Science (2016).

7) Turkin, A., van Oijen, A.M. & Turkin, A.A. Theory of bimolecular reactions in a solution with linear traps: Application to the problem of target search on DNA. Physical Review E 92, 052703 (2015).

8) Leith, J.S. et al. Sequence-dependent sliding kinetics of p53. Proc Natl Acad Sci U S A 109, 16552-7 (2012).

9) Halford, S.E. & Marko, J.F. How do site-specific DNA-binding proteins find their targets? Nucleic Acids Res 32, 3040-52 (2004).

10) Jin, T. et al. Structures of the HIN domain:DNA complexes reveal ligand binding and activation mechanisms of the AIM2 inflammasome and IFI16 receptor. Immunity 36, 561-71 (2012)